# Understanding ethnic inequalities in hearing health in the UK: a cross-sectional study of the link between language proficiency and performance on the Digit Triplet Test

Harry Taylor [1,2] Nick Shryane [1,2] Dharmi Kapadia [2,3] Piers Dawes [4,5] Paul Norman [6]

► Prepublication history and supplemental material for this paper are available online. To view these files, please visit the journal online (http://dx.doi.org/10.1136/bmjopen-2020-042571).

For numbered affiliations see end of article.

**Correspondence to**
Harry Taylor;
harry.taylor@manchester.ac.uk

## ABSTRACT

**Introduction** Research using the UK Biobank data has shown ethnic inequalities in hearing health; however, the hearing test used may exhibit a disadvantage for non-native language speakers.

**Objectives** To validate the results of the UK Biobank hearing test (Digit Triplet Test, DTT) against self-reported measures of hearing in the dataset and create classifications of hearing health. To observe if language proficiency and migration age have the same effect on hearing health classification as on the DTT in isolation. Our hypothesis is that language proficiency acts differently on the DTT, demonstrating that the DTT is biased for non-native speakers of English.

**Design** Latent classes representing profiles of hearing health were identified from the available hearing measures. Factors associated with class membership were tested using multinomial logistic regression models. Ethnicity was defined as (1) White, native English-speaking, (2) ethnic minority, arrived in the UK aged <12 or (3) ethnic minority, arrived aged >12.

**Participants** The UK Biobank participants with valid hearing test results and associated covariates (N=151 268).

**Outcome measures** DTT score, self-reported hearing difficulty, self-reported hearing difficulty in noise and hearing aid use.

**Results** Three classes of hearing health were found: 'normal', 'generally poor' and 'only subjectively poor'. In a model adjusting for known confounders of hearing loss, a poor or insufficient hearing test result was less likely for those with better language (OR 0.69, 95% CI 0.65 to 0.74) or numerical ability (OR 0.71, 95% CI 0.67 to 0.75) but more likely for those having migrated aged >12 (OR 3.85, 95% CI 3.64 to 4.07).

**Conclusions** The DTT showed evidence of bias, having greater dependence on language ability and migration age than other hearing indicators. Designers of future surveys and hearing screening applications may wish to consider the limitations of speech-in-noise tests in evaluating hearing acuity for populations that include non-native speakers.

## Strengths and limitations of this study

► To our knowledge, the first study comparing self-reported and psychophysical hearing outcomes for ethnic minorities using the UK Biobank data.

► Potential language bias in the UK Biobank's hearing test was investigated for the first time.

► All available hearing outcomes from the UK Biobank were combined into three latent classes; the behaviour of these classes was substantively consistent with the literature.

► This study uses the available language proficiency data in the UK Biobank, which is limited.

## INTRODUCTION

Hearing impairment is one of the most common chronic age-related health conditions[1]; it is estimated that one in five people in the UK will have hearing loss, characterised by 25 decibels (dB) or more hearing loss in the better ear, by 2035.[2] Ethnic inequalities in health outcomes are well documented and have been a key focus for the UK government over the last 25 years.[3] Yet research into ethnic inequalities in hearing health in the UK has been limited due to insufficient ethnic minority sample sizes.

This research study is concerned with the ethnic inequality observed by Dawes *et al* while investigating the correlates of hearing loss using the UK Biobank data.[4] The study found that people who reported a non-White ethnic background were 3.27 (95% CI 3.07 to 3.48) times more likely to have hearing loss than those who identified as White British. This result was consistent with reports of higher levels of other chronic health conditions among minority groups in the UK,[5] but contradicted observations from the

USA where hearing loss was 70% lower in Black groups compared with White groups.[6]

Hearing acuity in the UK Biobank is measured using the Digit Triplet Test (DTT), a psychophysical test of speech recognition that involves perception of three spoken digits. Other UK health surveys collect hearing data using methods based on pure-tone audiometry (PTA),[7 8] a non-language-based evaluation of hearing acuity employing detection of pure tones. The DTT generally shows good correlation with PTA measures[9]; however, the word-based approach of speech-in-noise tests measures language comprehension as well as hearing acuity. This could present a scenario whereby a poor test result for a non-native speaker is due not to poor hearing, but to a lower level of language proficiency compared with native speakers.[10] This language penalty has been shown to increase the more linguistically complex the speech-in-noise test.[11]

Given the potential conflation of language ability and hearing acuity in the DTT, the present study divides the ethnic minority cohort into those likely to have native-level English proficiency, and those who migrated to the UK as an adolescent or older. It is hypothesised that, correcting for respondents' self-reported hearing health and related factors, DTT scores for the older-migrating cohort will be worse due to their likely lower language proficiency.

The present research therefore aims to validate the results of the DTT against three self-reported measures of hearing provided in the UK Biobank. These measures will be used to create classifications of hearing health. We will see if language proficiency and migration age have the same effect on the hearing health classifications as for the DTT in isolation.

Our hypothesis is that the DTT is biased for non-native speakers of English. If the DTT is not biased, it would be expected that language test scores would have the same effect across every measure of hearing in the dataset. A different outcome would indicate a dependency on language ability. If, after controlling for language ability, the DTT still shows poorer outcomes for the ethnic minority groups, it can be concluded that inequalities are present regardless of any language bias.

## METHOD
### Sample and design
The study population was participants in the UK Biobank, a biomedical study of over 500 000 adults aged 40–69 years recruited in the UK from 2006 to 2010.[12] Participants were invited to the study based on their proximity to 1 of 25 test centres and being registered with the National Health Service.[12] The study design was cross-sectional, using only the first wave of the UK Biobank responses. The present research considers the 151 268 adults who produced valid DTT results and had a complete set of variables. We used the Strengthening the Reporting of Observational Studies in Epidemiology (STROBE) cross-sectional checklist when writing our report.[13]

## Measures
### Hearing health
Four measures of hearing health were used to form the hearing health classifications: the DTT, two self-reported measures of hearing difficulty and self-reported hearing aid use. All data, including the psychophysical measure, were collected in a self-administered manner via the touchscreen questionnaire undertaken in Biobank test centres.

As a speech-in-noise test, the DTT was devised to detect impaired speech intelligibility in noisy conditions,[14] which is one of the most frequently reported hearing difficulties associated with hearing loss.[15] The DTT was originally validated using participants whose first language was English.[16] The test produces a Speech Recognition Threshold (SRT) measured using signal-to-noise ratio in dB,[14] with lower scores representing better performance. The lowest score in the better ear was used in the present study, with scores above −5.5 dB characterised as 'insufficient/poor' as per Dawes *et al*.[4] Respondents who stated they were completely deaf or used a cochlear implant were excluded due to having no DTT data.

For the self-reported measures, respondents were asked: 'Do you have any difficulty with your hearing?' (*Yes/No/I am completely deaf /Do not know/Prefer not to answer),* and 'Do you find it difficult to follow a conversation if there is background noise (such as TV, radio, children playing)?' (*Yes/No/Do not know/Prefer not to answer*). The completely deaf category was amalgamated with the 'Yes' category. Respondents were asked if they currently use a hearing aid 'most of the time' (*Yes/No/Prefer not to answer*). Participants responding 'Do not know' or 'Prefer not to answer' to any of the hearing questions were excluded from the dataset.

### Correlates of hearing loss
Hearing health is affected by a number of demographic, socioeconomic and lifestyle factors, as well as comorbidity from associated diseases.[6] These factors were controlled for in the final analysis.

Hearing loss is more likely for men, and with increasing age.[17 18] Socioeconomic status has been shown to affect hearing, including education[19] and income.[20] Education was coded as either no formal qualification, secondary school-level qualifications or any further education. The Townsend Index, an area-level deprivation measure, was used instead of household income, which had a high number of missing values.

Exposure to noise is a well-established factor in hearing loss.[6] Measures were included for whether the participant had ever worked in an environment where they had to shout to be heard, or if they had previously been exposed to loud music.

Lifestyle factors known to affect hearing were included. Smoking[6] was categorised as never smoker or former/

current smoker. Alcohol consumption[21] was calculated based on estimated weekly intake per type of drink, and an ethanol content estimate for each type. Where an estimate was unavailable, a predicted value was assigned according to alcohol-intake frequency. The resulting intake was categorised against the 112 g/week recommended by the UK guidelines.[22]

Measures for secondary illnesses known to affect hearing were included. Hypertension was characterised as an average measurement of >140 mm Hg systolic or >90 mm Hg diastolic as per Agrawal *et al.*[6] Diabetes[6] and stroke[23] were based on a diagnosis from a doctor. Cardiovascular disease[6] was indicated by self-reported heart attack, stroke, angina, deep vein thrombosis or other related conditions. The use of medication for secondary conditions that may have ototoxic side effects[24] was also included.

The prevalence of disabling hearing loss is lower in high-income countries than elsewhere, partly due to hearing loss acquired during childhood.[25] As such, early life exposure (age <15 years) to the most common childhood diseases known to affect hearing was included in the model: meningitis,[26] measles, mumps and rubella.[27]

### Correction for bias

Language proficiency was evaluated according to two language-based questions included in the UK Biobank's Fluid Intelligence (FI) assessment: (1) 'Stop means the same as?' (*Pause/Close/Cease/Break/Rest*); (2) 'Bud is to Flower as Child is to?' (*Grow/Develop/Improve/Adult/Old*). Given the association between hearing loss and cognitive performance,[28] numerical questions from the FI test were also included to correct for general cognitive ability. Respondents who did not take the cognitive tests, or ran out of time answering them, were categorised separately.

The UK Biobank coded participants' ethnicity according to the 2001 Census ethnic groups. The present study was not concerned with differences between specific ethnic groups, but rather the risk that non-native English speakers may be penalised on the DTT. As such, participants were categorised according to likely language proficiency. Those who migrated to the UK aged under 11 years were said to have native proficiency, as per Nazroo's investigation of variations in mental illness rates according to language ability.[29] Given the focus on language, the 'White other' ethnic group was classified as an ethnic minority group, with the exception of those who were born in a majority English-speaking country (Australia, Canada, Channel Islands, Ireland, Isle of Man, New Zealand, the UK, the USA). Ethnicity was ultimately coded as (1) White, native English speaking (NE); (2) Black and Minority Ethnic (BME), native English speaking or early-migrating (NEEM) and (3) BME later-migrating.

Data collection for the UK Biobank generally took place in clinical research facilities or serviced office space; however, mobile assessment centres were used in Swansea and Wrexham.[30] Anecdotal evidence suggested that acoustic background noise conditions were worse in

the mobile units. Descriptive statistics (see online supplemental table 1) revealed poorer average speech recognition thresholds for these centres. As such, we corrected for test centre type.

### Analysis

A Latent Factor Analysis (LFA) was initially used to combine all four hearing health measures into a single latent dimension. This model exhibited unusual behaviour by way of inflated coefficients for factor variables. This unreliability suggested that latent hearing health was not well characterised by a single dimension; that is, self-reported hearing difficulties may not always correlate well with psychophysical measures of hearing acuity. This inconsistency could also have been due to external factors that act differently on each measure, for example, propensity to report health problems, or bias in the hearing test for certain groups.

Latent Class Analysis was employed to overcome the constraints of an LFA by allowing for the existence of arbitrarily different 'classes' of latent hearing health. A latent construct of hearing health was formed using the four hearing measures. The suitability of this construct as a proxy to understand the latent trait of hearing health was evaluated by studying its association with known correlates of hearing loss. Analyses were performed in MPlus V.8 using the three-step approach to correct for classification uncertainty in the latent classes.[31]

### Patient and public involvement

Although the UK Biobank participants were involved in the design of the biobank resource itself, neither they, or any member of the public, were involved in this secondary analysis.

## RESULTS
### Sample characteristics

A total of 165 306 participants provided valid hearing test data, reducing to 151 268 after the removal of cases having missing data (online supplemental figure 1). Table 1 shows the characteristics of the sample for each group in the analysis. Levels of 'poor' or 'insufficient' hearing according to the DTT are similar for the White NE and BME NEEM group; however, the later-migrating BME group had over twice the levels of hearing loss according to DTT performance. The two BME groups had similar levels of self-reported hearing difficulty.

### Latent variable analysis

A combination of exploratory and confirmatory approaches was used to determine the properties of the hearing health latent classifications. It was expected that there would be a particular group who exhibited insufficient or poor performance on the DTT, but did not self-report hearing problems or use a hearing aid. To test this, three-class and four-class confirmatory models were created with this group specified; other group thresholds

**Table 1** Sample characteristics (N=151 268); values are numbers (percentages)

| | White, native English speaking | BME, native English speaking or early-migrating | BME later-migrating |
|---|---|---|---|
| N | 134 563 | 5942 | 10 763 |
| **Age (years)** | | | |
| 40–44 | 12 634 (9.4) | 1399 (23.5) | 1751 (16.3) |
| 45–49 | 15 809 (11.7) | 1534 (25.8) | 1650 (15.3) |
| 50–54 | 19 136 (14.2) | 1249 (21.0) | 1776 (16.5) |
| 55–59 | 23 407 (17.4) | 798 (13.4) | 1914 (17.8) |
| 60–64 | 35 446 (26.3) | 606 (10.2) | 2010 (18.7) |
| 65+ | 28 131 (20.9) | 356 (6.0) | 1662 (15.4) |
| **Sex** | | | |
| Female | 72 161 (53.6) | 3382 (56.9) | 6091 (56.6) |
| Male | 62 402 (46.4) | 2560 (43.1) | 4672 (43.4) |
| Townsend deprivation (mean) | –1.4 | 0.3 | 0.9 |
| **Educational qualifications** | | | |
| CSE/GCSE/A-Level | 31 004 (23.0) | 1413 (23.8) | 2003 (18.6) |
| Degree/higher/prof qual | 82 369 (61.2) | 4043 (68.0) | 7202 (66.9) |
| No formal qualifications | 21 190 (15.7) | 486 (8.2) | 1558 (14.5) |
| **Ethnic group** | | | |
| White British or Irish | 133 808 (99.4) | 0 (0.0) | 0 (0.0) |
| White other | 755 (0.6) | 1915 (32.2) | 2943 (27.3) |
| Asian Bangladeshi | 0 (0.0) | 20 (0.3) | 46 (0.4) |
| Asian Indian | 0 (0.0) | 824 (13.9) | 2216 (20.6) |
| Asian other | 0 (0.0) | 93 (1.6) | 739 (6.9) |
| Asian Pakistani | 0 (0.0) | 206 (3.5) | 392 (3.6) |
| Chinese | 0 (0.0) | 70 (1.2) | 471 (4.4) |
| Black African | 0 (0.0) | 197 (3.3) | 1284 (11.9) |
| Black Caribbean | 0 (0.0) | 1396 (23.5) | 951 (8.8) |
| Black other | 0 (0.0) | 22 (0.4) | 30 (0.3) |
| Mixed | 0 (0.0) | 816 (13.7) | 318 (3.0) |
| Other | 0 (0.0) | 383 (6.4) | 1373 (12.8) |
| **Hearing measures** | | | |
| DTT: 'normal' | 119 257 (88.6) | 5331 (89.7) | 7251 (67.4) |
| DTT: 'insufficient' or 'poor' | 15 306 (11.4) | 611 (10.3) | 3512 (32.6) |
| Does not use hearing aid | 130 507 (97.0) | 5866 (98.7) | 10 566 (98.2) |
| Uses hearing aid | 4059 (3.0) | 76 (1.3) | 197 (1.8) |
| No self-reported hearing diff. | 96 400 (71.6) | 4779 (80.4) | 8822 (82.0) |
| Self-reported hearing diff. | 38 163 (28.4) | 1163 (19.6) | 1941 (18.0) |
| No self-reported hearing/diff. in noise | 85 757 (63.7) | 4274 (71.9) | 7362 (68.4) |
| Self-reported hearing/diff. in noise | 48 806 (36.3) | 1668 (28.1) | 3401 (31.6) |

Mixed group includes 'White and Black Caribbean', 'White and Black African', 'White and Asian' and 'other mixed'; other group includes 'Arab' and 'any other ethnic groups'.
BME, Black and Minority Ethnic; CSE, Certificate of Secondary Education; DTT, Digit Triplet Test; GCSE, General Certificate of Secondary Education.

were unconstrained, allowing the model to determine their properties. However, this class did not fit well with the data, having very low membership in both models.

Accordingly, an exploratory approach was revisited: unconstrained models with two to seven classes were created, and the model with best fit was selected. A

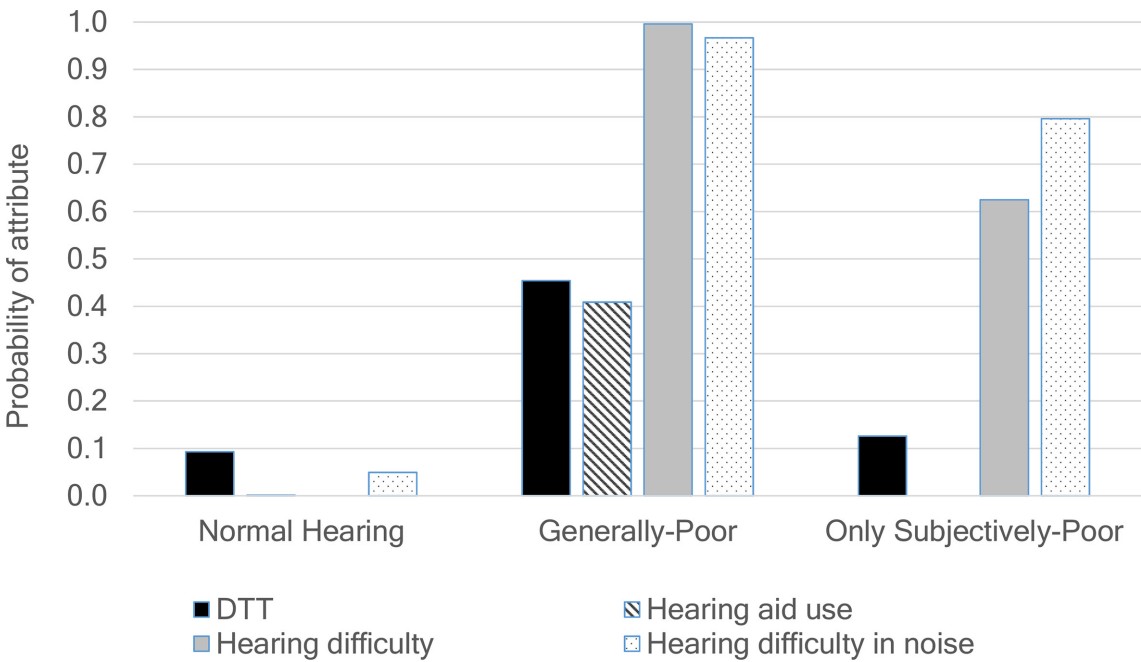

**Figure 1** Probabilities of three-class LCA model. DTT, Digit Triplet Test; LCA, Latent Class Analysis.

mixture of Bayesian Information Criterion (BIC) and Bootstrapped Likelihood Ratio Test (BLRT) is recommended for determining the optimum number of classes in a latent class analysis[32]; however, the BLRT models for four versus three classes were not reliable due to a non-positive definite first-order derivative product matrix; as such, the BIC alone was used to determine this number (online supplemental table 2).

### Hearing health classes
Figure 1 illustrates the hearing health classes produced in the three-class model, which were interpreted as follows:

'Normal' (prevalence 60.4%): characterised by low levels of hearing aid use, poor/insufficient hearing on the DTT or reporting hearing difficulty more generally.

'Generally poor' (prevalence 6.9%): those who reported hearing difficulties, the highest levels of poor/insufficient DTT performance and used hearing aids.

'Only subjectively poor' (prevalence 32.7%): those who reported hearing difficulties but had a low probability of having poor/insufficient hearing on the DTT.

### Latent class regression models
A multinomial logistic regression was performed using class membership as the dependent variable. The latent classes were regressed on a range of demographic, socioeconomic and health-related predictors, as well as known biases in the dataset. Uncertainty in the latent class classification was compensated for using the manual 'three-step' procedure.[31] Predictor variables were introduced hierarchically (see online supplemental files 4–6) to evaluate whether the ethnic inequalities could be explained by the introduction of particular factors.

The latent class regression models described above assumed complete mediation of the relationship between the observed predictor variables and the observed hearing health latent class indicators by the latent classes. This assumption did not tally with our hypotheses, which assumed that responses to the DTT in particular would be different for participants with different values on variables hypothesised to be related to language skills (ethnicity/migration history, language test score and numerical test score) and known biases (whether tested in a mobile centre). We therefore expanded the latent class regression models to include direct, non-mediated effects of these predictors on the DTT outcome in particular. These expanded models were similar in structure to psychometric models of (uniform) 'differential item functioning', with the difference that in psychometrics the latent mediator variable is usually a continuously distributed latent factor, whereas in our models it was a discrete latent class distribution. A non-zero estimate for these direct effects would indicate bias, whereby responses to the DTT variable were affected not only by hearing health, but also by other characteristics of the participant, such as language skill and ethnicity. Table 2 shows results for these key variables before and after these direct effects were introduced. A path diagram for the model can be seen in online supplemental figure 2.

Without the DTT direct effects, the BME NEEM group had significantly lower odds (OR 0.67, 95% CI 0.53 to 0.86) of being in the generally poor compared with the normal hearing group; however, these odds were no longer significant (OR 0.83, 95% CI 0.68 to 1.03) once direct effects were introduced. Higher numeric and language scores caused a significant reduction in the likelihood of having poor or insufficient hearing on the DTT, but did not have a significant impact on group membership. The later-migrating BME group were much more

**Table 2** Coefficients and 95% CI of key variables from multinomial logistic regression

| | Latent hearing health class | | |
| --- | --- | --- | --- |
| | **Only subjectively poor** | **Generally poor** | **Normal (ref)** |
| **Predictor** | **OR (95% CI)** | **OR (95% CI)** | **OR (95% CI)** |
| Excluding DTT direct effects | | | |
| BME (NEEM) | 0.86 (0.80 to 0.93)*** | 0.67 (0.53 to 0.86)** | 1.00 |
| BME, later-migrating | 0.94 (0.89 to 1.00) | 0.52 (0.44 to 0.62)*** | 1.00 |
| Language score: 1 correct | 1.01 (0.95 to 1.08) | 0.92 (0.81 to 1.05) | 1.00 |
| Language score: 2 correct | 1.05 (1.00 to 1.11) | 0.85 (0.77 to 0.95)** | 1.00 |
| Language score: not taken | 0.91 (0.81 to 1.03) | 0.86 (0.67 to 1.11) | 1.00 |
| Numeric score: >50% correct | 1.04 (1.00 to 1.09) | 1.02 (0.93 to 1.13) | 1.00 |
| Numeric score: not taken | 1.05 (0.94 to 1.17) | 1.22 (0.97 to 1.53) | 1.00 |
| Mobile test centre | 0.98 (0.88 to 1.09) | 1.05 (0.83 to 1.32) | 1.00 |
| Including DTT direct effects | | | |
| BME (NEEM) | 0.86 (0.80 to 0.93)*** | 0.83 (0.68 to 1.03) | 1.00 |
| BME, later-migrating | 0.94 (0.89 to 1.00) | 0.57 (0.49 to 0.67)*** | 1.00 |
| Language score: 1 correct | 1.00 (0.94 to 1.06) | 0.98 (0.87 to 1.10) | 1.00 |
| Language score: 2 correct | 1.04 (0.98 to 1.09) | 0.91 (0.83 to 1.00) | 1.00 |
| Language score: not taken | 0.89 (0.79 to 1.00) | 0.94 (0.76 to 1.17) | 1.00 |
| Numeric score: >50% correct | 1.04 (0.99 to 1.09) | 1.05 (0.96 to 1.15) | 1.00 |
| Numeric score: not taken | 1.08 (0.97 to 1.20) | 1.10 (0.90 to 1.35) | 1.00 |
| Mobile test centre | 1.00 (0.90 to 1.12) | 0.94 (0.76 to 1.17) | 1.00 |
| DTT direct effects | OR of having insufficient or poor hearing | | |
| BME (NEEM) | 1.04 (0.95 to 1.15) | | |
| BME, later-migrating | 3.85 (3.64 to 4.07)*** | | |
| Language score: 1 correct | 0.69 (0.65 to 0.74)*** | | |
| Language score: 2 correct | 0.61 (0.58 to 0.65)*** | | |
| Language score: not taken | 1.12 (1.00 to 1.26)* | | |
| Numeric score: >50% correct | 0.71 (0.67 to 0.75)*** | | |
| Numeric score: not taken | 1.34 (1.20 to 1.49)*** | | |
| Mobile test centre | 5.52 (4.99 to 6.10)*** | | |

All models controlled for age, sex, education, deprivation score, noise exposure, use of ototoxic medication, medical history, smoking and alcohol use, childhood illnesses and evaluation at a mobile test centre. Asterisk denotes significance: *95% interval; **99% interval; ***99.9% interval.
BME, Black and Minority Ethnic; DTT, Digit Triplet Test; NEEM, native English speaking or early-migrating.

likely than the White NE group to have insufficient or poor hearing (OR 3.85, 95% CI 3.64 to 4.07).

## DISCUSSION

This study further investigated the ethnic differences in hearing health observed by Dawes et al[4] by considering the heterogeneity in language ability of the ethnic minority group, and by validating the results of the DTT against other hearing health outcomes.

By dividing the ethnic minority cohort according to migration age, the present research reveals intergenerational differences in DTT outcomes. As hypothesised, it is the later-migrating BME ethnic minority group who bear the burden of poorer hearing outcomes: the UK-born/ early-migrating BME cohort had DTT results that were indistinguishable from the White NE cohort.

Aside from the DTT, the LFA was constructed using self-reported measures, which are thought to be reliable measures of hearing health. This construct indicates that ethnic minority groups are most likely to have normal hearing. However, even after correcting for hypothesised sources of bias, the results from the DTT suggest poorer hearing outcomes for the later-migrating BME group. This is consistent with literature showing ethnic inequalities in other physical and mental health outcomes,[3] which have been shown to have multiple causes, including socioeconomic and demographic factors, as well as stress caused by experiences of racial harassment and discrimination.[33]

The DTT showed evidence of language bias, having greater dependence on language ability and migration age than other hearing indicators. The self-report questions did not appear to demonstrate a strong linguistic bias, despite being posed in English. This may mean that the concept of hearing difficulties is reasonably consistently understood, and not prone to misinterpretation due to language barriers.

A secondary finding of this study regards mobile test centres, which caused the largest bias in the DTT, possibly due to elevated background noise levels in the test centres. Respondents who were tested at these centres were 5.52 times (95% CI 4.99 to 6.10) more likely to have poor/insufficient hearing on the DTT, but were not more likely to have any particular hearing classification. It is also of note that those who did not take the cognitive tests had worse DTT scores. It may be that a cognitive or language difficulty precluded participants from taking, or finishing, these tests.

The LFA produced classes that were substantively consistent with the literature. The likelihood of being in the generally poor hearing group was increased by exposure to noise and most of the health conditions known to affect hearing. The strongest indicator for the only subjectively poor hearing group was exposure to noise, which would appear consistent with a cohort who are concerned about their hearing. Although it may seem unusual that this cohort would perform well on a speech recognition test yet report hearing difficulties in daily life, such phenomena are not uncommon in the literature; explanations include audiometric notches,[34] cochlear synaptopathy[35] and health factors unique to the individual.[36]

One limitation to this study regards language measures: proficiency was determined by responses to just two questions, so may be imprecise. Despite this, the measures are still able to provide evidence for the hypothesis of DTT bias, as the LFA outcomes did not exhibit any dependency on language. It is important to note that the DTT is a test of sensorineural hearing loss, detecting only hearing loss of cochlear origin. The DTT, being a test of signal-to-noise ratio, allows conductive losses to be overcome by increasing the volume of the stimuli, which participants were permitted to do in the UK Biobank test. No other hearing tests were conducted; as such, the UK Biobank does not contain any data on conductive pathologies, otoscopy findings or middle ear function measures.

It must also be noted that the BME classes contain the 'White other' group, which may include participants from Western Europe who are not generally designated as minority ethnic. Regarding generalisability, the UK Biobank, while not employing a representative sample, is thought to be generalisable for disease-exposure relationships.[37]

The authors of the Biobank study did not explicitly state why the DTT was used over the pure-tone average method, but the available information points to practical factors, such the 'minimal impact on throughput' by incorporating the test with the existing touchscreen questionnaire,[38] and no requirement for calibrated equipment or a sound-treated test booth. Unfortunately, use of the DTT may have resulted in a test that is biased for non-native English speakers.

Even after accounting for this bias, the results suggest that there are still ethnic inequalities in hearing health, although with contradiction between psychophysical and self-reported outcomes. Designers of future studies must take precautions to ensure that this inequality can be reliably analysed by ensuring that hearing test measures are not language dependent. The present study may also have implications for hearing screening applications, which many countries currently implement using the DTT.[39] DTTs are being translated to an increasing number of languages,[40] although it may be some time before tests are available in all 6500 languages spoken globally. Until such time, non-linguistic hearing tests may be the most practical solution to lessen the dependency on language proficiency in psychophysical measures of hearing.

### Author affiliations

[1]Social Statistics, The University of Manchester School of Social Sciences, Manchester, UK
[2]School of Social Sciences, The University of Manchester Cathie Marsh Institute for Social Research, Manchester, UK
[3]Sociology, The University of Manchester School of Social Sciences, Manchester, UK
[4]Manchester Centre for Audiology and Deafness, The University of Manchester School of Health Sciences, Manchester, UK
[5]Manchester Academic Health Science Centre, Manchester University NHS Foundation Trust, Manchester, UK
[6]School of Geography, University of Leeds, Leeds, UK

**Acknowledgements** We would like to express our gratitude to Dr Antje Heinrich for her useful suggestions in response to a presentation on the initial stages of this work. We would also like to thank the BMJ Open editors and external reviewers for their thoughtful comments and suggestions.

**Contributors** HT carried out the conception and design of the study, analysis and interpretation of results, and drafting of the manuscript. NS, DK, PD and PN made substantial contributions to the conception and design of the study, and interpretation of data, as well as critically revising the manuscript. All authors approved the final version and agree to be accountable for all aspects of the work.

**Funding** This work was supported by the Economic and Social Research Council (grant number ES/P000665/1) and the NIHR Manchester Biomedical Research Centre.

**Competing interests** None declared.

**Patient consent for publication** Not required.

**Ethics approval** The UK Biobank received ethical approval from the National Health Service Research Ethics Service North West (11/NW/0382). Explicit consent for secondary analyses such as the one reported here, was obtained from participants in the UK Biobank.

**Provenance and peer review** Not commissioned; externally peer reviewed.

**Data availability statement** Data are available in a public, open access repository. The UK Biobank data are held in an open access resource available to researchers via the procedure described at https://ukbiobank.ac.uk/enable-your-research.

terminology, drug names and drug dosages), and is not responsible for any error and/or omissions arising from translation and adaptation or otherwise.

**ORCID iDs**
Harry Taylor http://orcid.org/0000-0002-0582-6851
Nick Shryane http://orcid.org/0000-0001-7963-4601
Dharmi Kapadia http://orcid.org/0000-0003-4007-1981
Piers Dawes http://orcid.org/0000-0003-3180-9884
Paul Norman http://orcid.org/0000-0002-6211-1625

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
