## [Reviewer comments · BMJ Open]

ARTICLE DETAILS

TITLE (PROVISIONAL)	Understanding ethnic inequalities in hearing health in the UK: a cross-sectional study of the link between language proficiency and performance on the Digit Triplet Test
AUTHORS	Taylor, Harry; Shryane, Nick; Kapadia, Dharmi; Dawes, Piers; Norman, Paul

VERSION 1 – REVIEW

REVIEWER	Christopher Spankovich University of Mississippi Medical Center, USA
REVIEW RETURNED	15-Sep-2020

GENERAL COMMENTS	bmjopen-2020-042571 presents an analysis of UK biobank data relative to the relationship between native language and hearing outcome measures. In general, the manuscript is very well-written and the analysis appears appropriate. I have but a few minor suggestions. 1. I recommend changing the title to something more specific in regards to relationship between native language and hearing status.2. Page 5 line 30 mentions objective measures, none the measures used are objective in nature, they all require a response from the patient and are thus subjective. Same thing on page 13 Line 17. Perhaps psychophysical measure vs self-reported.3. Is there any data on conductive pathologies, otoscopy findings, middle ear function measures? If so this would be a worthy additional consideration, if not, I would include as a limitation.
---

REVIEWER	Sumit Dhar Northwestern University, USA
REVIEW RETURNED	01-Oct-2020

GENERAL COMMENTS	The manuscript under review uses a large data set to demonstrate a language bias in the Digit Triplet Test -- a standard with growing popularity in diagnosing hearing loss with very demands on test condition and stimulus calibration. The bias, as predicted by the authors, is evident in a significant over estimation of hearing loss in Black or other minority individuals who have migrated when or after turning 12 years old. That language comprehension plays a role in influencing test results where material such as words or sentences are used as stimuli is well established. These results are especially significant because those findings are extended to the use of digits and a test which was built with the expectation to overcome such limitations. The manuscript is well-written and easy to follow. The only (very) minor suggestion I have is to be explicit that choosing "Do not
--

	know” Or “Prefer not to answer” options for any of the questions led to exclusion at the bottom of page 5. Of course, the conclusions are entirely dependent on the use of what appear to be state-of-the-art statistical techniques. I am not an expert statistician. So the editor is urged to seek the opinion of a domain expert. Failing this endorsement the quality of the entire manuscript will be in doubt. It is interesting that the estimation of hearing loss based on self report in response to questions posed in English does not demonstrate a linguistic bias. Perhaps a little more commentary about whether this should be considered a paradox may make the manuscript more interesting. Overall I enjoyed reading the manuscript
--	--

REVIEWER	Alexander Miething Department of Public Health Sciences, Stockholm University, Sweden
REVIEW RETURNED	26-Oct-2020

GENERAL COMMENTS	Statistical review of the manuscript „Understanding hearing health inequalities in ethnic minority groups in the UK” In general, the statistical methods applied in this study seem to sound. I admit that I had some difficulties to understand the design of Table 2, specifically how the “DTT direct effects” were obtained: were these estimates directly obtained from the multinomial regression model or from a separate model? I suggest that the authors expand the description of Table 2 in the section “Latent class regression models” on page 10 and provide more details on how the regression analysis was performed.
--

VERSION 1 – AUTHOR RESPONSE

Reviewers' Reports:

We greatly appreciate the reviewers' positive comments. We have carefully revised the manuscript according to those comments, and present our point-by-point response.

Reviewer 1: I recommend changing the title to something more specific in regards to relationship between native language and hearing status.

Authors: We appreciate the reviewer's recommendation. We have changed the title to “Understanding ethnic inequalities in hearing health in the UK: a cross-sectional study of the link between language proficiency and performance on the Digit Triplet Test”

Reviewer 1: Page 5 line 30 mentions objective measures, none the measures used are objective in nature, they all require a response from the patient and are thus subjective. Same thing on page 13 Line 17. Perhaps psychophysical measure vs self-reported.

Authors: We appreciate the reviewer's recommendation. We have changed each instance of "objective" to "psychophysical" and each instance of "subjective" to "self-reported".

Reviewer 1: Is there any data on conductive pathologies, otoscopy findings, middle ear function measures? If so this would be a worthy additional consideration, if not, I would include as a limitation.

Authors: The DTT is a test of sensorineural hearing loss, detecting only hearing loss of cochlear origin. The DTT, being a test of signal to noise ratio, allows conductive losses to be overcome by increasing the volume of the stimuli, which participants were permitted to do in the UK Biobank test. No other hearing tests were conducted; as such, UK Biobank does not contain any data on conductive pathologies, otoscopy findings, or middle ear function measures. We have added a statement to this effect to the manuscript text.

Reviewer 2: The only (very) minor suggestion I have is to be explicit that choosing "Do not know" Or "Prefer not to answer" options for any of the questions led to exclusion at the bottom of page 5.

Authors: We appreciate the reviewer's suggestion and have altered the text accordingly.

Reviewer 2: It is interesting that the estimation of hearing loss based on self report in response to questions posed in English does not demonstrate a linguistic bias. Perhaps a little more commentary about whether this should be considered a paradox may make the manuscript more interesting.

Authors: We appreciate the reviewer's suggestion for further commentary, and have added the following text to the manuscript: *"The DTT showed evidence of language bias, having greater dependence upon language ability and migration age than other hearing indicators. The self-report questions did not appear to demonstrate a strong linguistic bias, despite being posed in English. This may mean that the concept of hearing difficulties is reasonably consistently understood, and not prone to misinterpretation due to language barriers."*

Reviewer 3: I admit that I had some difficulties to understand the design of Table 2, specifically how the "DTT direct effects" were obtained: were these estimates directly obtained from the multinomial regression model or from a separate model? I suggest that the authors expand the description of Table 2 in the section "Latent class regression models" on page 10 and provide more details on how the regression analysis was performed.

Authors: We appreciate the reviewer's suggestion and have revised the manuscript accordingly to add some clarification. We also added a path diagram for the model, which can be seen in supplementary figure 2. The revision is as follows: *"The latent class regression models described above assumed complete mediation of the relationship between the observed predictor variables and the observed*

hearing health latent class indicators by the latent classes. This assumption did not tally with our hypotheses, which assumed that responses to the DTT in particular would be different for participants with different values on variables hypothesised to be related to language skills (ethnicity/migration history, language test score and numerical test score) and known biases (whether tested in a mobile centre). We therefore expanded the latent class regression models to include direct, non-mediated effects of these predictors on the DTT outcome in particular. These expanded models were similar in structure to psychometric models of (uniform) “differential item functioning”, with the difference that in psychometrics the latent mediator variable is usually a continuously distributed latent factor, whereas in our models it was a discrete latent class distribution. A non-zero estimate for these direct effects would indicate bias, whereby responses to the DTT variable were affected not only by hearing health, but also by other characteristics of the participant, such as language skill and ethnicity.”